# GameTraversalBenchmark: Evaluating Planning Abilities Of Large Language Models Through Traversing 2D Game Maps

**Muhammad Umair Nasir**
University of the Witwatersrand
Johannesburg, South Africa
muhammad.nasir@wits.ac.za

**Steven James**
University of the Witwatersrand
Johannesburg, South Africa
steven.james@wits.ac.za

**Julian Togelius**
New York University
New York, USA
julian@togelius.com

## Abstract

Large language models (LLMs) have recently demonstrated great success in generating and understanding natural language. While they have also shown potential beyond the domain of natural language, it remains an open question as to what extent and in which way these LLMs can plan. We investigate their planning capabilities by proposing GameTraversalBenchmark (GTB), a benchmark consisting of diverse 2D grid-based game maps. An LLM succeeds if it can traverse through given objectives, with a minimum number of steps and a minimum number of generation errors. We evaluate a number of LLMs on GTB and found that GPT-4-Turbo achieved the highest score of $44.97\%$ on GTB_Score (GTBS), a composite score that combines the three above criteria. Furthermore, we preliminarily test large reasoning models, namely o1, which scores $67.84\%$ on GTBS, indicating that the benchmark remains challenging for current models. Code, data, and documentation are available at https://github.com/umair-nasir14/Game-Traversal-Benchmark.

## 1 Introduction

Large language models, built atop the transformer architecture [1], are widely influential in the field of natural language processing [2], and have shown great promise in a variety of applications that were not originally envisioned as target domains, thereby, hinting towards more general Artificial Intelligence models [3, 4]. In recent years, numerous LLMs have emerged, each competing for state-of-the-art performance by continuously expanding the limits of their capabilities [5]. As a result, the accurate evaluation of LLMs has become a key focus for researchers [6–8].

LLMs are trained to predict the next token based on their current context, but an ongoing debate is whether these next-token predictors are capable of planning [9, 10]. Planning may include tasks in games [11, 12], robot control [13], questioning answering [14], or visual programming [15]. While these works show the capabilities of LLMs to plan, multiple complementary planning benchmarks are required to know whether these task-specific examples of planning are a common LLM ability. Therefore, we present a benchmark designed to evaluate the planning abilities of LLMs from a novel perspective. This benchmark serves as a complement to existing LLM assessments, focusing on tasks that are unlikely to be extensively represented in the training datasets of current LLMs. While prior

38th Conference on Neural Information Processing Systems (NeurIPS 2024) Track on Datasets and Benchmarks.

evaluations of planning abilities have been conducted in natural language environments [16, 17], our work introduces a task represented as a string of characters that depicts a 2D grid-based map. This representation is infrequently encountered by LLMs during training, preventing them from merely performing a lookup to obtain answers; instead, they must engage in effective planning to determine the optimal path.

Previous works have shown that LLMs can generate 2D grid-based maps with fine-tuning and training [18–20], and without [21, 12]. Consequently, the question of whether LLMs can successfully create 2D grid-based maps has been addressed. However, while an LLM can generate sequences of characters devoid of natural language meaning, we seek to investigate whether it can effectively plan using these same character sequences. This inquiry is the focus of our research.

A natural question arises: Why evaluate LLMs on a 2D map? The rationale is that a 2D map, represented as a sequence of characters, is interpretable by LLMs. Given adequate instruction, LLMs should therefore be capable of processing and comprehending the map's state, which encompasses the map itself, the agent's position, and the locations of the objectives. Previous studies have demonstrated that LLMs can process instructions that are out-of-distribution from their training data. Therefore, when provided with the map's state and the available actions for moving the agent, an LLM capable of planning should be able to generate a sequence of actions to reach the target. To generate the correct action sequence, the LLM must plan each move. Similar to how humans remain aware of their surroundings while traversing a path, an LLM needs to continuously observe the environment to produce the correct actions. But do LLMs actually observe and plan as they generate action sequences? To answer this, we introduce the `GameTraversalBenchmark` (GTB), which consists of a dataset of diverse maps.

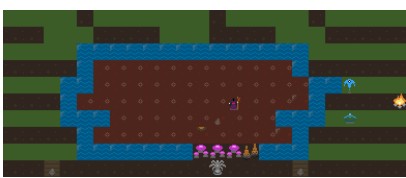

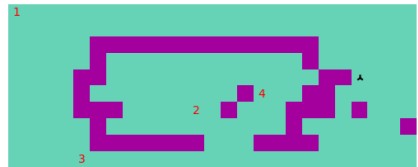

Figure 1: Top: An example of a level produced by the Word2World [12] algorithm and rendered using its tileset. Middle: Binary mask of the above map for visualising walkable tiles (teal), non-walkable tiles (purple), and the position of objectives (numbered). Bottom: Character representation of the map provided as input to the LLM.

In the following sections, we begin by describing the dataset collection process and the creation of GTB. Next, we evaluate several LLMs on the GTB benchmark, demonstrating that while the state-of-the-art GPT-4 achieves the highest performance, it still falls short of reaching 50%. We conclude with a discussion on how traversal problems can be made more challenging to encourage the research community to improve LLMs' planning capabilities.

## 2  GameTraversalBenchmark

In this section, we will discuss how we curated the dataset and the `GameTraversalBenchmark` (GTB). The dataset is generated via the pipeline introduced by our previous work, named as Word2World [12]. Word2World is an LLM-based game design system that creates a story and narrative for the game by extracting useful information, such as tileset and character descriptions, and goals for the player. After extracting useful information, Word2World first places alphanumeric characters for the environment of the world in a 2D grid-based map. After that, it places alphanumeric characters for the game characters, such as the protagonist and antagonist of the story, and the interactive object tiles needed for the protagonist to play the game. Then Word2World generates the coordinates of objectives needed to complete the game. The setting of the environment and game

characters is iteratively refined to generate more coherent maps. These worlds are generated using GPT-4-Turbo, GPT-4, GPT-3.5, Claude-3-Opus and Claude-3-Haiku.

We extract 150 of these maps, an example of which is illustrated by Figure 1. Each of the objectives for the LLM is to reach a certain coordinate and then to reach another coordinate, which serves as the next objective. These maps make good environments for judging the traversal abilities of an agent, as they have diverse patterns and different sizes. While there are datasets available from previous games [22, 23], GTB provides many different sized levels with deceptive paths to traverse.

## 2.1 Benchmark

We consider a generated map as one data point for the benchmark. The map's generated objective coordinates are the points where the LLM agent needs to traverse to attain the most rewards. Since we are interested in observing an LLM's planning ability from a traversal agent perspective, we compute the optimal path using $A^*$ [24] search, which is set to find the optimal (shortest) path within 25000 iterations.

The goal of the LLM agent is to traverse the world through these objectives. The LLM agent is required to generate a sequence of actions that should take the agent from current position to the position of the objective. Therefore, LLM agent can generate a sequence of actions for the current objective once. All generated actions in the sequence are then rolled out in simulation and the position of the LLM agent decides the rewards it gets. Table 1 shows how rewards are distributed. This new position of the LLM agent is the starting positon for the next objective. The LLM agent also needs to find the shortest path to each objective. Thus, an LLM agent receives the highest rewards for achieving the objective while minimising the number of actions and making the fewest errors while generating the solution. Such errors include incorrect actions generated by the LLM, failure to adhere to imposed constraints, or the generation of syntactically incorrect outputs.

Table 1: Reward distribution for an LLM agent.

| Rewards allowed | Reason | Reward |
|---|---|---|
| At each step | For each action taken. | -1 |
| | For each generation mistake. | -1 |
| At the end of each objective | 8 or more tiles away from the objective. | -100 |
| | 5 - 8 tiles away. | -50 |
| | 3 - 5 tiles away. | +25 |
| | 2 - 3 tiles away. | +50 |
| | 1 tile away. | +100 |
| | Exactly on the coordinate. | +200 |

Thus the maximum and minimum achievable reward per level is:

$$R_{max}^{(m)} = \sum_{m=0}^{M} (200 - A_{PL}^{*(m)}) \tag{1}$$

and

$$R_{min}^{(m)} = \sum_{m=0}^{M} (-100 - A_{PL^m}^{*(m)} - \varepsilon_{max}^{(m)}), \tag{2}$$

where 200 is the maximum reward obtained by reaching the exact coordinate of the objective, $-100$ is the minimum reward for being farthest away from the coordinate, and $A_{PL^m}^*$ is the path length of the objective calculated by an $A^*$ agent. $\varepsilon_{max}^{(m)}$ is the maximum number of generation errors possible during the level $m$, and $M$ is the total number of 2D game map in the dataset. Thus, we have the following as the normalised and scaled reward calculations across the dataset, which is also called GTB_Score:

$$GTB\_Score = \frac{1}{M} \sum_{m=0}^{M} \left( \frac{(R^{(m)} - LLM_{PL}^{(m)} - \varepsilon^{(m)}) - R_{min}^{(m)}}{R_{max}^{(m)} - R_{min}^{(m)}} \right), \tag{3}$$

where $M$ is the number of levels in the dataset.

$R^{(m)}$ is determined by reaching a specific coordinate in the level and querying Table 1 for the corresponding reward, which is based on the distance to the goal. $LLM_{PL}^{(m)}$ is the path length the LLM-agent took to reach the coordinates across all objectives in the level $m$. $\varepsilon^{(m)}$ is the negative reward for the total error given to the LLM agent for making an error while generating. $R_{min}^{(m)}$ and $R_{max}^{(m)}$ are the minimum and maximum possible rewards for the level $m$, respectively. Thus, this reward function takes into account how close the LLM agent has reached to the exact coordinate of the objective, how many steps it took, and if it made any errors.

The negative reward for error is placed to judge the LLM by its capability to control the generation. We give LLMs 10 attempts per objective and record the errors they make before succeeding in generating the right actions. The errors could be a wrong action generated or a syntax error. Errors are recorded as:

$$MGE = \frac{1}{M} \sum_{m=0}^{M} \varepsilon_{GE}^{(m)}, \tag{4}$$

where $MGE$ represents the average number of errors generated $\varepsilon_{GE}^{(m)}$ for the level $m$, across the whole dataset.

Similarly,

$$MPL = \frac{1}{M} \sum_{m=0}^{M} PL^{(m)} \tag{5}$$

$$MAT = \frac{1}{M} \sum_{m=0}^{M} AT^{(m)} \tag{6}$$

In Equation 5, we have $PL_m$ as the path length for the map $m$. We do not add it to the path length if the agent moves up and then moves down, or vice versa, and moves left and moves right, or vice versa, as the position of the agent stays the same. This calculates the *mean path length* (MPL). For all the actions, including actions that are not included in MPL, we calculate them through Equation 6 to find *mean actions taken* (MAT), where we count the length of actions taken $AT_m$ by an LLM for the map $m$.

Therefore, GTB provides the following metrics for evaluations:

1. **GTB_Score**: Score that is calculated by Equation 3. This is the score that determines the place of an LLM on the scoreboard.

2. **MGE**: Score that indicates how many errors occurred across the dataset.

3. **MPL**: Score that indicates the path length taken by the agent.

4. **MAT**: Score that indicates total actions taken by the agent.

5. **Top-0 Accuracy**: Score that indicates how many times the agent has reached the exact coordinate of the objective. The score is calculated by:

$$\frac{\text{Number of objective coordinates reached}}{\text{Total number of objectives}} \tag{7}$$

6. **Top-1 Accuracy**: Score that indicates how many times the agent has reached the 1-tile window of the objective. Reaching objective coordinates is not counted in this score. The score is calculated by:

$$\frac{\text{Number of 1-tile window to objective coordinates reached}}{\text{Total number of objectives}} \tag{8}$$

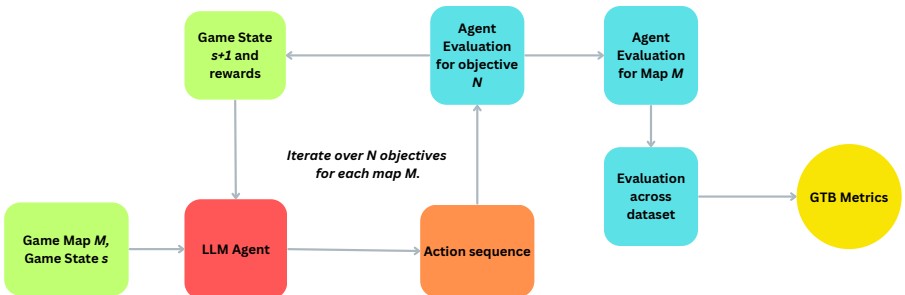

Figure 2: An illustration of the GTB evaluation loop. Each game map $M$ is evaluated turn-by-turn for all objectives $N$ present in it. A game state $S$ includes the game map, the position of LLM agent, the position of objectives, and current rewards. The updated state, $S+1$, has the updated position of the LLM agent in it and updated rewards. LLM agent produces a sequence of actions for that particular objective and is evaluated for that objective. Once all objectives are iterated over, the agent evaluation is stored and the loop moves to the next map. Once all maps are evaluated, the GTB metrics are calculated.

7. **Top-5 Accuracy**: Score that indicates how many times the agent has reached 2 - 5 tiles away from the objective coordinate. Reaching the coordinate and 1-tile window is not counted in the score. The score is calculated by:

$$\frac{\text{Number of 5-tile window to objective coordinates reached}}{\text{Total number of objectives}} \tag{9}$$

Figure 2 shows how GTB metrics are generated. A game map $M$ is picked from the dataset and the current state of the map $s$ is given to the LLM agent which includes the map, the position of the agent, and the position of the objective. Some more constant information is given to the LLM, which includes what each tiles refers to in the map, walkable tiles, and important tiles. This loop is iterated over $N$ objectives of the map. The LLM agent produces the action sequence which is required for the agent to traverse from the agent's position to the objective's position. Then the state is updated by the current position of the agent and the next objective position. This time the rewards for the previous objectives are also provided. Figure 3 illustrates how exactly the input is provided to the LLM, while Figure 4 illustrates how we prompt an LLM in GTB.

GTB provides zero and one-shot evaluations for all the above metrics. One-shot evaluation is provided for researchers who are interested in few-shot evaluations. Primarily, we are interested in evaluating LLMs that may achieve high scores on GTB via zero-shot. Here, zero-shot means that the LLM agent will have to generate action sequences in one go. If it makes a generation error, it is not given the previous actions to re-generate the actions. For one-shot, we give the LLM one extra chance per objective in each game map, if it has not reached the exact coordinate of the objective. In this case, the LLM receives the actions it took in the previous shot, the distance between the objective and the final position, and the reward it received.

## 2.2 Exploring the Benchmark

We now show the diversity of the dataset that GTB evaluates the LLM agents on. We first consider the diverse map sizes contained in the dataset. We observe in Figure 5a that there is a vast distribution of map sizes in the dataset. If we look into $width \times height$ of the different maps in the dataset, we see that we have small maps $5 \times 10$ size and a cluster around the size $8 \times 15$. The majority of the distribution lies in between the width of 20 to 40 to a height of 10 to 20. The distribution extends with a considerable amount of maps with heights of more than 20 and around 35, while a good spread of maps has widths up to just below 100. These map sizes express diversity, allowing other factors, like a number of objectives and optimal path lengths, to be more diverse.

Figure 5c also adds to the diversity of the dataset. While we have more maps with optimal paths around 20 to 30 actions, but we have a maximum of just below 120 actions and a minimum of 5. This also shows diversity in the difficulty of traversing the maps. The dataset includes simpler maps, solvable by an $A^*$ agent in 5 to 10 actions, which provide LLMs with an opportunity to achieve high

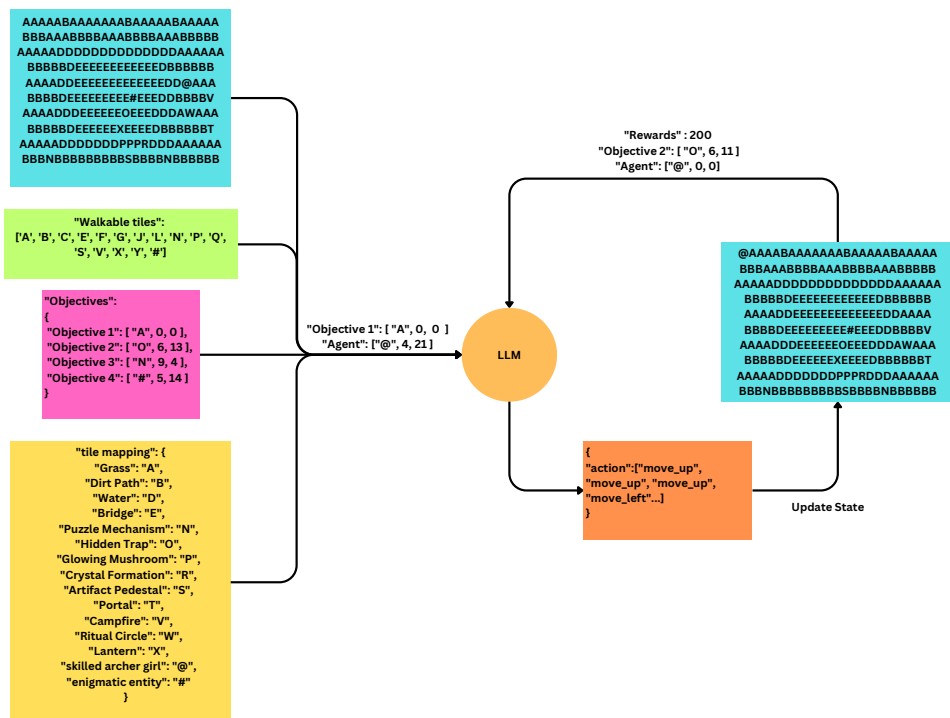

Figure 3: Illustrates an example of an input to the LLM and the output of the action sequence for the objective.

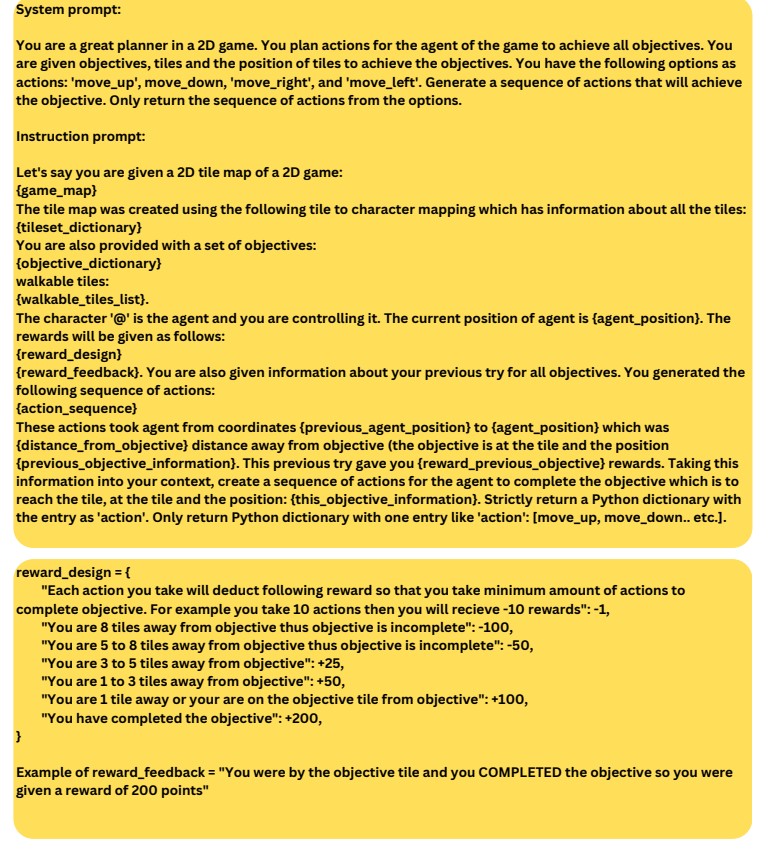

Figure 4: Example of a prompt in GTB to generate actions.

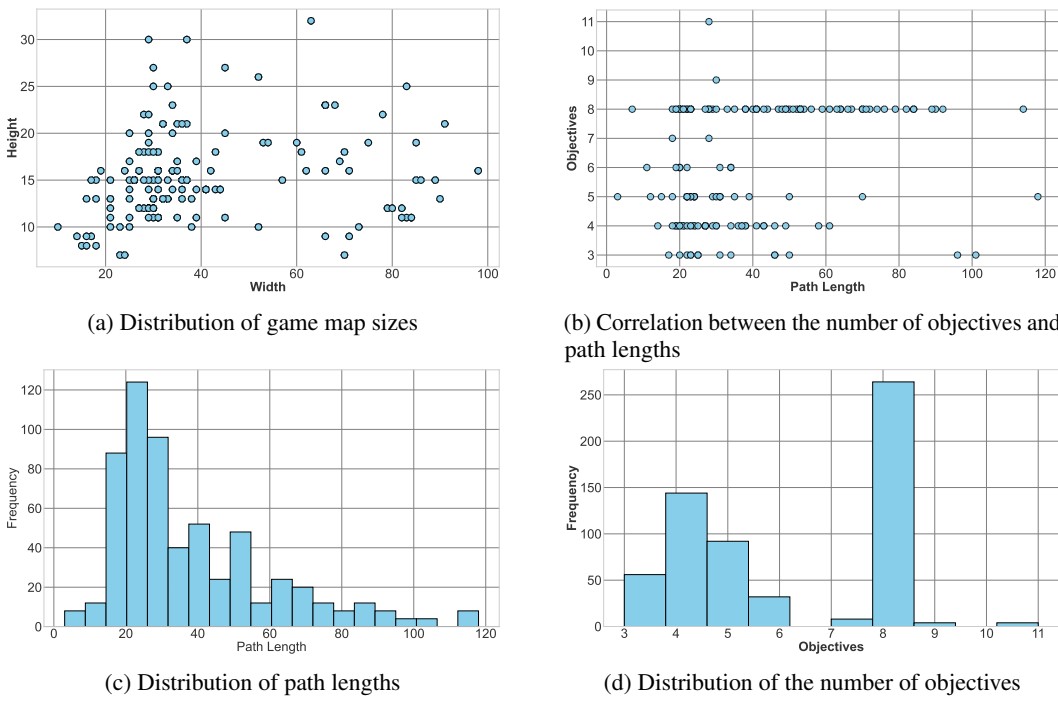

(a) Distribution of game map sizes

(b) Correlation between the number of objectives and path lengths

(c) Distribution of path lengths

(d) Distribution of the number of objectives

Figure 5: Exploring `GTB` in terms of path lengths and number of objectives.

rewards. However, true planning capabilities are better assessed on maps where the optimal paths exceed 100 steps.

Lastly, we show the distribution of the number of objectives in Figure 5d across the dataset. We observe a sufficient amount of diversity by having a distribution spread from 3 to 8 objectives but we also have a map with 11 objectives. This distribution, along with path length from Figure 5c, also indicates a difficult traversal dataset as we have more than 60 maps with 8 objectives. These span across maps with different path lengths as we have a nice spread of distribution in Figure 5a and 5c. Thus, smaller maps with 8 objectives will also be challenging for LLMs, similar to larger maps with 8 objectives. We also look into the correlation of the number of objectives and path lengths of the maps in Figure 5b. We can observe that the correlation is not high which makes the benchmark more interesting. We have path length around 100 with only 3 objectives while a map with 11 objectives has a path lenth of around 30. This distribution makes it interesting as there is no set rule for the LLM to follow which may lead to a plan for all maps. The LLM may need to generate very large number of actions per objective and very small number of actions as well.

## 3 Baseline Results

To show the relevance of `GTB` in the current state-of-the-art LLMs, we evaluate a few families of LLMs. Table 2 shows results for different LLMs. The best-performing LLMs, such as *GPT-4-Turbo* and *Claude-3-Opus*, are among the state-of-the-art LLMs as well when it comes to benchmarks for natural language understanding tasks, code generation tasks, as well as on PlanBench [16]. Thus, the results are consistent with other benchmarks, making `GTB` a uniques option to evaluate planning in LLMs. These results demonstrate that the benchmark is a tough challenge for LLMs and there is a big room for improvement in the planning abilities of today's LLMs.

Furthermore, we can observe from Table 2 that all LLMs except the first 4 LLMs are worse than a *Random-FP* agent. The Random-FP agent computes the difference in distance between two objectives, using this value to determine the length of the required action sequence. Then it produces that many actions uniformly at random. MGE, MPL and MAT are not applicable in this case as there will be no generation error and the path length is fixed. Questions therefore arise as to whether LLMs with lesser capacity are just randomly generating sequences of actions. Observation of the behaviour exhibited

Table 2: Results of some LLMs on GTB. The maximum GTBS is 100. Results are the mean over 3 seeds and a standard deviation is also shown. We observe that state-of-the-art LLMs do not achieve high scores on GTB. Random-FP is a random baseline with a fixed length of action sequences that are calculated by the difference between the agent and the objective position. Random-RP is a random baseline that generates a random length of action sequences.

| Model | GTBS↑ | MGE↓ | MPL↓ | MAT↓ | Top-0 Acc.↑ | Top-1 Acc.↑ | Top-5 Acc.↑ |
|---|---|---|---|---|---|---|---|
| GPT-4-Turbo | 44.97 ± 0.22 | 0.03 ± 0.01 | 80.91 ± 0.69 | 80.97 ± 0.62 | 19.2 ± 0.24 | 17.66 ± 0.46 | 23.05 ± 1.03 |
| GPT-4-o | 30.95 ± 0.76 | 0.53 ± 0.06 | 85.42 ± 0.58 | 85.91 ± 0.61 | 7.84 ± 0.17 | 11.34 ± 0.36 | 18.99 ± 0.45 |
| Claude-3-Opus | 28.65 ± 0.59 | 0.02 ± 0.01 | 100.41 ± 0.37 | 100.44 ± 0.36 | 5.49 ± 0.65 | 12.35 ± 0.43 | 22.72 ± 0.09 |
| Claude-3-Sonnet | 18.54 ± 0.22 | 0.0 ± 0.0 | 75.05 ± 0.38 | 76.22 ± 0.41 | 0.73 ± 0.05 | 3.80 ± 0.11 | 13.64 ± 0.31 |
| Random-FP | 18.02 ± 0.50 | N/A | N/A | N/A | 0.91 ± 0.13 | 2.77 ± 0.09 | 12.41 ± 0.27 |
| Gemma-7B | 15.65 ± 0.21 | 0.11 ± 0.02 | 37.06 ± 0.55 | 40.16 ± 0.71 | 0.29 ± 0.05 | 2.05 ± 0.05 | 10.95 ± 1.41 |
| GPT-3.5-Turbo | 14.34 ± 0.31 | 16.49 ± 0.39 | 46.83 ± 0.69 | 48.23 ± 0.91 | 0.44 ± 0.23 | 3.49 ± 0.34 | 11.88 ± 0.31 |
| LLaMa-3-8B | 14.08 ± 0.67 | 0.54 ± 0.06 | 55.38 ± 1.19 | 56.06 ± 1.12 | 0.21 ± 0.00 | 2.27 ± 0.32 | 8.54 ± 0.53 |
| LLaMa-3-70B | 11.39 ± 1.36 | 0.41 ± 0.04 | 266.88 ± 23.79 | 267.02 ± 23.88 | 1.06 ± 0.05 | 4.84 ± 0.7 | 16.63 ± 0.96 |
| Claude-3-Haiku | 10.81 ± 1.15 | 0.0 ± 0.0 | 69.14 ± 8.18 | 70.09 ± 8.21 | 0.09 ± 0.06 | 1.25 ± 0.88 | 7.32 ± 1.96 |
| Mixtral-8×7B | 9.35 ± 0.56 | 9.61 ± 0.41 | 152.73 ± 5.43 | 152.99 ± 5.46 | 0.67 ± 0.24 | 2.85 ± 0.16 | 10.19 ± 0.19 |
| Random-RP | 3.04 ± 0.65 | N/A | 278.48 ± 0.67 | 297.22 ± 1.02 | 0.0 ± 0.0 | 0.20 ± 0.18 | 15.37 ± 2.26 |

by the various LLMs suggests that the actions taken by LLMs other than GPT-4 and Claude-3-Opus do not appear meaningful, and could very well be random. The most likely explanation for this is that these LLMs are unable to internally construct a map or representation of the environment's state. A truly random agent, *Random-RP* generates random action sequence of random lengths and achieves a GTBS of 3, implying that generating the right sequence length is not a random task—it needs to be planned. LLMs performing better than Random-RP agent also means the LLM may have a sense of how long the path should be, but were not able to produce the right actions. This suggests that a less capable LLM might grasp the relationship between the start and target positions of the objective, but fails to comprehend the path planning process required to navigate between them.

Table 3: Results for selected LLMs on one-shot GTB.

| Model | GTBS↑ | MGE↓ | MPL↓ | MAT↓ | Top-0 Acc.↑ | Top-1 Acc.↑ | Top-5 Acc.↑ |
|---|---|---|---|---|---|---|---|
| GPT-4-Turbo | 52.07 ± 0.25 | 0.02 ± 0.01 | 73.88 ± 0.71 | 73.91 ± 0.7 | 22.01 ± 0.19 | 17.83 ± 0.61 | 24.95 ± 2.01 |
| LLaMa-3-70B | 15.17 ± 1.53 | 0.22 ± 0.07 | 213.28 ± 24.17 | 213.35 ± 24.21 | 2.16 ± 0.07 | 6.67 ± 0.52 | 19.93 ± 0.34 |
| Gemma-7B | 13.63 ± 0.31 | 0.07 ± 0.01 | 35.16 ± 0.51 | 35.41 ± 0.61 | 0.18 ± 0.12 | 2.02 ± 0.09 | 15.95 ± 0.91 |

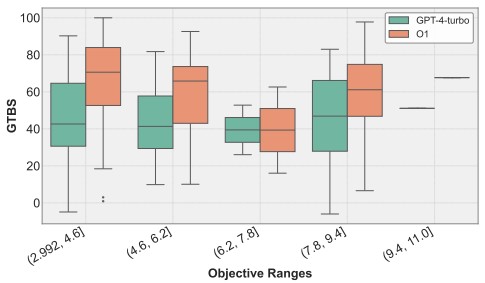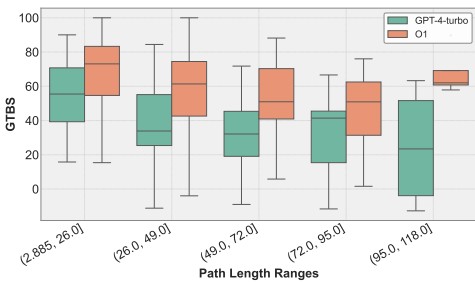

(a) Comparison on number of objectives.

(b) Comparison on path lengths.

Figure 6: Comparison of o1 and GPT-4-Turbo

Furthermore, it is noteworthy that smaller LLMs tend to produce shorter path lengths, possibly recognising the need to generate fewer steps. However, they often fail to produce the correct actions. LLaMa-3-70B and Mixtral-8×7 (a 56B parameter model) traverse more than the lesser capacity models. These two LLMs reach more objectives but explore more while reaching them. Gemma-7B, LLaMa-7B, and Claude-3-Haiku, the lower capacity LLMs, reach the objectives fewer times than the above-mentioned LLMs, but also take fewer actions to reach them. Therefore, they score better.

For one-shot GTB we observe that GPT-4-Turbo and LLaMa-3-70B improve, while Gemma-7B performs worse than zero-shot. A possible reason might be how much attention a lesser capable model can give to its context, while better models can give better attention to all of its contexts. This behaviour is also noticed in PlanBench [16].

### 3.1 Preliminary Results On Large Reasoning Models

We further test GTB on o1 [25], a recent large reasoning model (LRM), to see how the current state-of-the-art performs on GTB. The underlying LLM of o1 is trained using reinforcement learning to curate its outputs through a private Chain-of-Thought reasoning process [26], which enhances its ability to simulate a "thinking" phase before generating responses. Furthermore, o1 is a reasoning model, meaning that the inference stage, which contains many more steps than the typical LLM, is computationally and monetarily expensive. For this reason, it cannot be straightforwardly compared with other LLMs - it is fundamentally a different kind of model. For the same reason, we only include preliminary results based on a single run in this paper. In the single run, we found that the model outperformed the top-performing LLM on GTB. However, despite surpassing GPT-4-Turbo, it was unable to exceed a score of 70 on GTBS, as demonstrated in Table 4.

Table 4: Results of o1 and o1-mini. GPT-4-Turbo added for reference.

| Model | GTBS↑ | MGE↓ | MPL↓ | MAT↓ | Top-0 Acc.↑ | Top-1 Acc.↑ | Top-5 Acc.↑ |
|---|---|---|---|---|---|---|---|
| O1 | 67.84 | 0.12 | 51.35 | 51.73 | 50 | 10.76 | 13.19 |
| O1-mini | 61.36 | 0.83 | 82.83 | 82.95 | 46.98 | 6.70 | 14.38 |
| GPT-4-Turbo | 44.97 ± 0.22 | 0.03 ± 0.01 | 80.91 ± 0.69 | 80.97 ± 0.62 | 19.2 ± 0.24 | 17.66 ± 0.46 | 23.05 ± 1.03 |

Figure 6 presents a comparison between o1 and the best-performing seed from GPT-4-Turbo in terms of the number of objectives and path lengths. While o1 outperforms GPT-4-Turbo across all ranges of both metrics, the observed differences are relatively modest. As shown in Table 4, although o1 achieves twice the Top-0 accuracy of GPT-4-Turbo, the latter produces fewer generation errors, as indicated by MGE, and demonstrates significantly higher Top-1 and Top-5 accuracy. Nonetheless, o1 consistently identifies shorter paths compared to GPT-4-Turbo. These results suggest that while LRMs perform well on GTB, there remains considerable room for improvement.

# 4    Related Work

Conventionally, LLMs are evaluated for their natural language understanding and generation abilities. In this context, Hendrycks et al. [6] introduced the Massive Multitask Language Understanding (MMLU) benchmark which consists of 57 tasks spanning subjects such as elementary mathematics, computer science, and law. Similarly, Beyond the Imitation Game benchmark (BIG-bench) [7] was also introduced to evaluate LLMs across 204 diverse tasks. Other evaluations, such as Chatbot Arena [27] and MT-Bench [8], focus on measuring the general capabilities of chatbots. While these benchmarks assess overall performance across multiple tasks, there are numerous specialised benchmarks that evaluate specific downstream tasks. For example, SocKET [28] assesses social knowledge, TRUSTGPT [29] focuses on ethics, MATH [30] evaluates mathematical problem-solving, APPS [30] and HumanEval [31] test code generation abilities, FreshQA [32] and TRUTHFULQA [33] examine question-answering capabilities, and SafetyBench [34] evaluates the safety performance of LLMs.

Benchmarks specifically designed to evaluate the planning abilities of LLMs are also related to GTB. API-Bank [35] evaluates an LLM on the ability to continuously plan, retrieve, and call multiple APIs. The most closely related works are PlanBench [16] and AutoPlanBench [17] which uses Planning Domain Description Language (PDDL) [36] and convert them into natural language to evaluate the planning abilities of LLMs. While previous works have demonstrated the capacity of LLMs to play games [12, 37–40] to the best of our knowledge, no prior work has focused on evaluating LLMs specifically within the context of games.

# 5    Limitations

Although GTB has provided valuable insights into the planning abilities of LLMs, it is important to acknowledge its limitations, which, if addressed, could further advance the field. The 2D game maps are static, which is still challenging but can be increased in difficulty, such as moving non-player characters, enemies that may attack, tiles that may end in terminating conditions etc. Furthermore, GTB focuses solely on pathfinding abilities, which, while critical, represent only one dimension of planning. The action space is also limited to only 4 discrete actions. GTB has a static prompt that evaluates all the LLMs, which may be easier for larger LLMs to follow instructions then smaller ones.

# 6    Conclusion And Future Directions

This research introduces GameTraversalBenchmark (GTB), a novel benchmark for evaluating LLMs planning abilities through traversal in a 2D grid-based map. We provide extensive metrics to give insights towards planning abilities in LLMs. We then evaluate many LLMs on GTB_Score. We also evaluate LRMs on GTB_Score. We believe that there is a lot of room in this direction for research as LLMs have not been evaluated as traversal agents before but have been used in previous research as game-playing abilities. For future work, we would like to see how LLMs of all scales perform when they are fine-tuned on GTB. We would like to see how much smaller LLMs can improve upon fine-tuning, and after fine-tuning, how much such fine-tuning will generalise. Once we have LLMs that can perform well on GTB, we would like to extend the work by letting LLMs generate the state representation as well. It would also be valuable to have a prompt generation mechanics that can create prompts for all LLMs separately. A mechanism introduced by [41] could be useful to enhance LLMs on GTB. We hope that our contribution will push the boundaries of current state-of-the-art LLMs planning abilities as they are not able to achieve high scores on GTB, yet.

# 7    Acknowledgments

We would like to acknowledge our peers at Game Innovation Lab, New York University, USA for their support as funding and fellow members for advice during experimentation. We would also like to acknowledge the Robotics and Autonomous Intelligence Lab, University of the Witwatersrand, South Africa for their advice during experimentation. We would also like to acknowledge that there are no negative societal impacts of our work, to the best of our knowledge, even though it involves

pre-trained LLMs but as these LLMs were not trained by us therefore there are no ethical concerns that need to be addressed.

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
