# OpenReview forum: "GameTraversalBenchmark: Evaluating Planning Abilities Of Large Language Models Through Traversing 2D Game Maps"
_NeurIPS.cc/2024/Datasets_and_Benchmarks_Track — NeurIPS 2024 Track Datasets and Benchmarks Poster_

### Official Review · Reviewer_GU9P · 2024-07-12
**Review on Game-Traversal-Benchmark**

**Rating:** 7
**Confidence:** 3

**Review:**

This paper presents a benchmark for evaluating the planning ability of LLMs in 2D-grid traverse games, namely the GTB. The topic is interesting. The benchmark seems generally well designed. The presentation is fine. However, I do think there are some vital questions about the premise of this benchmark and some detailed settings that need to be answered. Overall, I rate this work a 6, with room for further discussion.

**Strengths:**

1. Planning has been a recent intersection of LLM and reinforcement learning, which facilitates applying LLMs to real-world problems involving an agent interacting with its environment. This work proposes a benchmark to evaluate LLMs’ ability to plan. This topic is definitely of interest to the deep-learning community.

2. The GTB benchmark proposes evaluating LLMs in 2D-grid environments. This seems smart, as 2D-grid maps can generally be fed into all LLMs by transferring to lines of text.

3. The GTB benchmark appears to be very difficult for current state-of-the-art LLMs, leaving plenty of room for future advances.

4. The authors are clearly familiar with related works and grasp current and future research directions for LLMs.

**Additional Feedback:**

None

**Clarity:**

The presentation of this paper is clear enough to convey the key information of this work, yet I found there is a number of elementary mistakes in the writing, which is a little disappointing for NeurIPS standard:
1. There should be a dot at the end of line 40, i.e. “…is understandable by LLMs. Hence, if…”

2. Line 98, “the maximum achievable reward” -> “the maximum and minimum achievable reward”

3. Maybe line 111 should be “gamma is set to 1” or “we set gamma to 1”?

4. I don’t understand why the columns in Figure 7 are not aligned with the axes.

5. The caption of Figure 7 is mistaken, should be “the number of objectives” rather than “lengths of optimal paths”

6. The A* agent in line 160 gave me confusion at first. Please unify the denotation of AStar and A*.

**Correctness:**

The evaluation methods and experimental design are overall appropriate, with some questions to be discussed (given above).

**Documentation:**

Yes, there is sufficient detail to support reproducibility.

**Ethics:**

No, I do not suspect there are any ethical concerns with the submission that warrant further discussion or review.

**Limitations:**

My biggest concern with this work is whether the 2D traverse games written in the symbolic text are an appropriate environment to evaluate the planning ability of LLMs. As the paper demonstrates, only 3 of the 10 LLMs tested actually outperformed a random baseline. But I don’t think the rest 7 LLMs doesn’t has ANY ability to plan (or only have negative planning ability). Maybe their planning ability needs to be mined in a different way. Maybe graphic text converted directly from 2D maps is not suitable or comprehensive for evaluating LLMs. After all, this kind of text is out-of-distribution for the LLMs. I would like to see a discussion on this in the revision.

**Opportunities For Improvement:**

1. I don’t understand how R_min from equation (2) is the minimum possible reward for an agent. If AStar_PLn is the length of the optimal (shortest) path, then wouldn’t a sub-optimal path that leads to 8 tiles away from the objective result in a smaller “minimum reward”? Or is it just a baseline to subtract from the actual reward obtained by the agent in equation (3)?

2. What does the “generation errors” particularly refer to? An elaboration on this can be helpful.

3. “We do not add it to the path length if the agent moves up and then moves down, or vice versa, and moves left and moves right, or vice versa, as the position of the agent stays the same.” What about when the agent walks a circle? Does it still counts as 0 path length? Anyway, I don’t think it is reasonable to reduce the agent’s path length like this. A path that involves meaningless moves like moving up then down is the result of sub-optimal planning, so all steps from the agent should be taken into account.

**Relation To Prior Work:**

Yes, the relationship between this work and previous contributions is well discussed.

**Summary And Contributions:**

Recent advances in Large Language Models (LLMs) has shown potential of planning with LLMs. The authors propose Game-Traversal-Benchmark (GTB), which is a benchmark consisting diverse 2D grid-based traverse game maps to evaluate the planning and reasoning abilities of an LLM. The GTB benchmark evaluates a LLM by 3 criteria including: 1. if the LLM can guide the agent to the objectives; 2. the number of agent’s steps; 3. the number of LLM’s generation errors. The authors also evaluate state-of-the-art LLMs on GTB to demonstrate the challenge in the benchmark and its potential to evoke future advances on improving the planning ability of LLMs.

---

> ### Author Rebuttal · Authors · 2024-08-13
>
> Good day, Reviewer GU9P.
>
> Thank you very much for your kind review! We hope that our work contributes to the community the way you have described.
>
> The following is our try to address the issues and insights you mentioned:
>
> ### Opportunities For Improvement
>
> **1: I don’t understand how R_min from equation (2) is the minimum possible reward for an agent. If AStar_PLn is the length of the optimal (shortest) path, then wouldn’t a sub-optimal path that leads to 8 tiles away from the objective result in a smaller “minimum reward”? Or is it just a baseline to subtract from the actual reward obtained by the agent in equation (3)? What does the “generation errors” particularly refer to? An elaboration on this can be helpful.**
>
> **Reply:**
>
> This is certainly a baseline to be subtracted from the reward obtained
> by the agent, as shown in equation (3). A minimum reward can be less than
> R_min which will result in a negative GTB_Score. However, this has a very low
> probability because to achieve this the agent has to produce a path length less
> than the optimal path which becomes difficult to do so because the agent has to explore
> to reach the objective coordinates. Figures b and c, in the PDF attached, show that GPT-4-Turbo has scored in the negative only 4 times out of 151.
>
> **2: What does the “generation errors” particularly refer to? An elaboration
> on this can be helpful.**
>
> **Reply:** Generation errors include all errors related to an LLM. For ex-
> ample, sometimes an LLM will produce an action that is not valid, such as
> ”move_right_then_move_up” which will create an error. Sometimes an LLM will
> not be able to produce the right answer. All of these are added in generation
> errors. We have elaborated in the revised PDF as well.
>
> **3: “We do not add it to the path length if the agent moves up and then
> moves down, or vice versa, and moves left and moves right, or vice versa, as the
> position of the agent stays the same.” What about when the agent walks a circle?
> Does it still counts as 0 path length? Anyway, I don’t think it is reasonable to
> reduce the agent’s path length like this. A path that involves meaningless moves
> like moving up then down is the result of sub-optimal planning, so all steps from
> the agent should be taken into account.**
>
> **Reply:** When the agent moves in a circle then it is counted within the path even though the displacement is 0.
> We see and agree that the all meaningless moves should be counted. We have updated this and run all LLMs with this update. Attached PDF has two tables with Mean Action Taken (MAT). These will be added as columns in Table 1 and 2, respectively, of the paper.
>
> ### Limitations
>
> **My biggest concern with this work is whether the 2D traverse games written
> in the symbolic text are an appropriate environment to evaluate the planning
> ability of LLMs. As the paper demonstrates, only 3 of the 10 LLMs tested
> actually outperformed a random baseline. But I don’t think the rest 7 LLMs
> doesn’t has ANY ability to plan (or only have negative planning ability). Maybe
> their planning ability needs to be mined in a different way. Maybe graphic text
> converted directly from 2D maps is not suitable or comprehensive for evaluating
> LLMs. After all, this kind of text is out-of-distribution for the LLMs. I would
> like to see a discussion on this in the revision.**
>
> **Reply:** As 4 out of 10 LLMs (we made a mistake here, 4 out of 10 perform better
> and we wrote 3 out of 10) perform better than the Random-FP baseline, we give
> a reason what might have happened in section 3, Baseline Results, paragraph
> 3. Here, we propose that smaller LLMs might capture the fact that they need
> to produce shorter paths but eventually they could not produce more accurate
> paths. On the other hand, LLaMa-3-70B and Mixtral8×7B were ranked below
> smaller LLMs but their accuracies are higher than smaller ones, which tells us
> that they took their time exploring and eventually reached the objective. This
> is also true for top-ranked LLMs but they were also much accurate.
>
> Furthermore, in future works we have now added a discussion on how we can make it such
> that the planning abilities for all the LLMs are captured in the benchmark.
> Which may be possible with a more sophisticated method of prompting, like
> Quality Diversity based rainbow method [1]. Also, we have added a new section of Limitations where we have mentioned that GTB has a static prompt which may be biased towards certain LLMs based on how they are trained.
>
>
> ### Clarity
>
>
> We apologize that we made naive mistakes and we thank you for mentioning
> these mistakes. We have fixed these mistakes as you have directed.
>
>
> Lastly, we want to thank you for your great review. We have made all the changes to the paper as mentioned above while staying in the page limit. This will be reflected in the final version.
>
>
> We believe that your reviews surely made the paper much better for the community. We hope that our rebuttals have satisfied your questions and alleviated your concerns!
>
>
>
> With best regards,
>
> Authors of the paper
>
> ### References:
>
> [1] Samvelyan, M., Raparthy, S.C., Lupu, A., Hambro, E., Markosyan, A.H.,
> Bhatt, M., Mao, Y., Jiang, M., Parker-Holder, J., Foerster, J. and Rockt ̈aschel,
> T., 2024. Rainbow teaming: Open-ended generation of diverse adversarial
> prompts. arXiv preprint arXiv:2402.16822.

---

> > ### Comment · Reviewer_GU9P · 2024-08-15
> >
> > I sincerely appreciate the detailed reply from the authors, which answers most of my concerns. Given the obvious improvement in this revision, I happily raise my rating to 7.

---

> > > ### Author Response · Authors · 2024-08-15
> > > **Thank you very much!**
> > >
> > > We thank our Reviewer GU9P for the great suggestions which will surely create further impact in this paper. We are happy to include them and we are highly grateful to our Reviewer as they increased their rating of the paper!

---

### Official Review · Reviewer_52Ke · 2024-07-25
**The paper introduces 2D Game Maps based traversal benchmark to evaluate planning abilities of LLMs**

**Rating:** 7
**Confidence:** 4
**Correctness:** Yes, the paper misses some ablation s…
**Clarity:** Yes, some improvements of paper writi…

**Review:**

- Utility of benchmark: GTB benchmark will be quite useful to test the planning abilities of LLMs. It is not evident what is the real-world use case of the benchmark. Can it be used to train LLMs to create a better planner? It will be useful to discuss regarding this in the paper.

- Ablation studies: Figures 5, 6 and 7 shows different statistics of the dataset. It will be worth to show the performance of baseline (for eg. GPT-4-Turbo) different settings of optimal path lengths, objectives, size of 2D game maps, and others.

- Paper writing could be improved. It would be good to address the limitations of the dataset and benchmark, structuring the figures, minor typo on captions of figure 6 and 7.

**Strengths:**

- The paper introduces Game-Traversal-Benchmark (GTB), a novel benchmark for evaluating LLMs planning abilities through traversal in a 2D grid-based map.

- The authors provide extensive metrics to give insights towards planning abilities in LLMs. Further, they evaluate many LLMs on GTB.

**Additional Feedback:**

None

**Documentation:**

Yes

**Limitations:**

I urge the authors to include the limitations of the benchmark and acknowledge the societal impact of their work. I believe that the benchmark does not seem to have negative societal impact of the work. The limitations could be that
- 2D game maps are static
- Benchmark test only path finding capability of LLMs with limited set of action space.

**Opportunities For Improvement:**

Discussed in the reviews section.

**Relation To Prior Work:**

Yes

**Summary And Contributions:**

- The paper proposes Game-Traversal-Benchmark (GTB), a benchmark consisting of diverse 2D grid-based game maps to evaluate the planning and reasoning abilities of an LLM.
- The authors provide statistics regarding the GTB dataset and benchmark and further provide several metrics for evaluations: GTB score, MGE, top-k accuracy etc.
- Extensive experiments are conducted using various LLMs.

---

> ### Author Rebuttal · Authors · 2024-08-13
>
> Good day, Reviewer 52Ke,
>
> Thank you very much for your kind review!
>
> The following is our try to address justified issues or insights you mentioned:
>
> **Utility of benchmark: GTB benchmark will be quite useful to test the planning abilities of LLMs. It is not evident what is the real-world use case of the benchmark. Can it be used to train LLMs to create a better planner? It will be useful to discuss regarding this in the paper.**
>
> **Reply:** We have made space the the paper to discuss this topic. We added that as the scores progress on GTB, we can say that LLMs now have a procedure to build a world model inside them. Right now, LLMs can take a guess and the larger ones may have the understanding of how to move between two coordinates, avoiding characters that are unwalkable, to some extent. Furthermore, we added in the future directions that we are interested to investigate LLMs fine-tuned on GTB and observe if smaller LLMs can improve after the fine-tuning. Furthermore, we added that we would also like to see how such tuning would generalize.
>
> **Ablation studies: Figures 5, 6 and 7 shows different statistics of the dataset. It will be worth to show the performance of baseline (for eg. GPT-4-Turbo) different settings of optimal path lengths, objectives, size of 2D game maps, and others.**
>
> We have added two figures that depicts correlation between:
> 1) the number of objectives and GPT-4-Turbo scores on GTB_S
> 2) the path lengths and GPT-4-Turbo scores on GTB_S
>
> Normalised Scaled Rewards here are GTB_S.
>
> **Paper writing could be improved. It would be good to address the limitations
> of the dataset and benchmark, structuring the figures, minor typo on captions
> of figure 6 and 7.**
>
> We have polished the writing in the revised edition. We have added a Limitations section as you advised and we have corrected figure 6 and 7. Furthermore, we have added an extra figure that depicts the correlation between path lengths and objectives.
>
> We have attached PDF to show the additions. Furthermore, we have made the space to add these within the 9 page limit.
>
> **Limitations**
>
> We have added a limitation section that states the following:
>
> Even though GTB has provided useful insights into the planning abilities of LLMs, we still like to address the limitations of GTB which can progress the field, if surpassed. The 2D game maps are static, which is still challenging but can be increased in difficulty by having more dynamics in it. GTB is made to test only the path finding ability of LLMs, which is important, but only one aspect of planning. The action space is also limited to only $4$ discreet actions. GTB has a static prompt that evaluates all the LLMs, which may be easier for larger LLMs to follow instructions then smaller ones.
>
> **Acknowledgements**
>
> We have added the societal impact in the acknowledgements section, which is as follows:
>
> We would like to acknowledge our peers at Anonymous Lab, anonymous university for their immense support and advice during experimentation. We would also like to acknowledge that there are no negative societal impacts of our work, to the best of our knowledge, even though it involves pre-trained LLMs but as these LLMs were not trained by us therefore there is no ethical concerns that are needed to be handled.
>
> Thank you again for your review! We believe that it has already improved the paper a lot.
>
> With best regards,
> Authors of the paper

---

> > ### Comment · Reviewer_52Ke · 2024-08-26
> >
> > Thanking the authors for detailed response.
> >
> > Overall, I feel good with the paper and would improve my ratings to 7.

---

> > > ### Author Response · Authors · 2024-08-26
> > > **Thank you very much!**
> > >
> > > We thank our reviewer 52Ke for increasing their score and providing valuable reviews that have improved the paper for the community!
> > >
> > > Thank you very much!

---

### Official Review · Reviewer_vLVW · 2024-07-25
**Interesting benchmark for LLMs but without source codes, supplementary materials, or documentation...**

**Rating:** 6
**Confidence:** 4
**Clarity:** the paper is well written

**Review:**

Using non-language-related task to evaluate LLMs in GTB benchmark is important and interesting to understand the reasoning capacities of LLMs. The results of the evaluation have revealed some important characteristics of existing popular LLMs in general planning and reasoning capabilities beyond language-generation tasks.

However, the submitted paper lacks of documentation, proper links, supplementary materials, codes, or datasets, making it questionable for correctness, truthfulness, and reproducibility.

**Strengths:**

Using non-language-related task to evaluate LLMs in GTB benchmark is important and interesting to understand the reasoning capacities of LLMs. The results of the evaluation have revealed some important characteristics of existing popular LLMs in general planning and reasoning capabilities beyond language-generation tasks.

**Additional Feedback:**

all suggestions comments are given above

**Correctness:**

The design of the experiment seems reasonable and appropriate, but without proper documentation, source code, or datasets, it's difficult to ensure its correctness.

**Documentation:**

there's no documentation, URL, datasets, source code, or supplementary available with this paper submission. There's no support for reproducibility.

**Ethics:**

there's no ethical concerns in this paper

**Limitations:**

Lack of documentation, data, codes, or link to the running benchmark is the main limitation of the paper.

**Opportunities For Improvement:**

more materials like documentation, datasets, links, or source code can be provided to strengthen the paper.

**Relation To Prior Work:**

the related work is clear

**Summary And Contributions:**

The paper presents a new benchmark for evaluating reasoning capabilities of LLMs called Game-Tree-Benchmark (GTB). Unlike other benchmarks for LLMs reasoning, GTB uses path traversal/navigation planning problem in a grid-world environment. The state of the grid-navigation-world and the navigation task objectives are translated into textual representation to be fed as parts of the prompt to the LLM, so that the LLM can provide the output as the plan of traversing the path in the environment. Evaluation using popular LLMs showed interesting and important characteristics of planning capabilities of different LLMs beyond language-based tasks.

---

> ### Author Rebuttal · Authors · 2024-08-13
>
> Good day, Reviewer vLVW,
>
> Thank you very much for your kind review!
>
> We apologize for not providing documentation, code, and the dataset earlier. We have no excuse that can justify it, truly we lost the race with time. We have now provided documentation, code and dataset at https://anonymous.4open.science/r/Game-Traversal-Benchmark-5D47/README.md. Once we get accepted, we will provide the GitHub link in the abstract of the paper. Currently, the documentation is hosted on the repo but we plan to shift it to GitHub Pages in the future.
>
> There is no justification for our mistake but we hope that this makes up for it. We hope that this strengthens the paper and alleviates your concerns!
>
> Please let us know if there are any other concerns we can tackle.
>
> With best regards,
> Authors of the paper

---

### Author Rebuttal · Authors · 2024-08-13

Good day to all our Reviewers, ACs, SACs, and PCs

Thank you very much for your kind reviews!

We appreciate immensely that you have taken out time and made the efforts to give these review. These reviews will definitely improve the paper. We have attended to all of the concerns reviewers raised and we are attaching a PDF of extra results that we managed to compile. The PDF contains the addition of Mean Action Taken (MAT) for tables 1 and 2 in the paper. These will be added as columns. We have also added three new figures. The first figure explores the benchmark more by illustrating the correlation between the number of objectives and the path lengths. Second and third figures depict the GPT-4-Turbo score with respect to path lengths and the number of objects.

Furthermore, we will reply to each review in-depth, separately. I hope our efforts will satisfy reviewers and alleviates their concerns.

Thank you again!

With warm regards,
Authors

---

### Decision · Program_Chairs · 2024-09-26

**Decision:**

Accept (Poster)

**Comment:**

This paper sets up a benchmark for path-planning problem with reward collection, and wants to pose it to evaluate the planning abilities of LLMs. The way they do this is to convert a visual grid into an alphabetic (ASCII) map--with different letters having different meanings (passable vs. impassable; whether there are rewards to be collected etc). It is this alphabetic map they give to LLMs to see if they can plan within them.

The paper reviewed well, with rebuttals and some responses.

As a researcher with background in planning myself, I found the benchmark to be frankly quite quixotic. It seems that the simple path planning problem is deliberately being obfuscated just to show LLMs will fail on them. This is something that is brought up by the third reviewer. It seems to me that in the era of VLMs, this particular benchmark--without the obfuscation--can be given directly to the VLMs.